# Evolutionary Reward Design and Optimization with Multimodal Large Language Models

Ali Emre Narin

Kabatas Erkek High School

aliemre2024@gmail.com

## Abstract

*Designing effective reward functions is a pivotal yet challenging task for Reinforcement Learning (RL) practices, often demanding domain expertise and substantial effort. Recent studies have explored the utilization of Large Language Models (LLMs) to generate reward functions via evolutionary search techniques [7]. However, these approaches overlook the potential of multimodal information, such as images and videos. In particular, prior methods predominantly rely on numerical feedback from the RL environment for doing evolution, neglecting the incorporation of visual data that could be obtained during training. This study introduces a novel approach by employing Multimodal Large Language Models (MLLMs) to craft reward functions tailored for various RL tasks. The methodology involves providing MLLM with the RL environment's code alongside its image as context and task information to generate reward candidates. Then, the chosen agent undergoes training, and the numerical feedback from the environment, along with the recorded video of the top-performing policy, is provided as feedback to the MLLM. By employing an iterative feedback mechanism through evolutionary search, MLLM consistently refines the reward function to maximize accuracy. Testing on two different agents across two distinct tasks points to the preeminence of our approach over previous methodology, which themselves outperformed 83% [7] of reward functions designed by human experts.*

## 1. Introduction

Large Language Models (LLMs) have shown remarkable success in distinct tasks. State-of-the-art models such as Gemini [2], Palm [4], and GPT-4 [10] have achieved results comparable to human experts on different benchmarks. In this paper, we are specifically interested in their capabilities in designing Reward functions for Reinforcement Learning practices. Recent studies have shown that GPT-4 can autonomously generate reward functions for multiple agents in IsaacGYM by taking the environment code as context and employing evolutionary search [7]. Impressively, it achieved results similar to and sometimes even better than those of human experts.

This result is very important for two reasons: firstly, the task of designing effective reward functions is notoriously challenging and time-consuming, and this approach streamlines the process by creating an end-to-end pipeline; secondly, by requiring no additional task-specific modifications, it showcases the generalization capabilities of evolutionary search on reward design.

However, a significant shortcoming of this approach, and LLMs in general, is that they can only operate on textual and numerical data. In contrast, when designing reinforcement learning strategies, human experts often leverage visual data to gain a deeper understanding of the problems that can be solved and improvements that can be made. It is our hypothesis that incorporating visual data could provide the model with enhanced comprehension, thus leading to improved accuracy.

We introduce EROM: "**E**volutionary **R**eward Design and **O**ptimization with **M**ultimodal Large Language Models (MLLMs)" method as a novel way to generate reward functions. In the EROM method, we utilize MLLMs' zero-shot coding abilities to generate reward functions. First, we provide the MLLM with the environment as context by providing the source code; then, we give it the description of the task, guidelines for reward function generation, and an image of the idle agent. After it generates the first iteration of the reward function, we provide feedback from the environment both numerically and visually by providing the video of the agent . Using evolutionary search, it generates a better set of reward functions, and this process iteratively continues.

Our contributions with the EROM method are as follows:
1. To the best of our knowledge, this is the first work that tests the MLLMs' abilities on reward function generation using evolutionary search.
2. We show that capturing the video (or image of an idle agent) of the top-performing policy and providing it to

the MLLMs as feedback helps the performance, compared to just providing textual reflection.

3. By enhancing the qualities of an autonomous method that outperformed 83% human experts, we contribute to the advancement of autonomous reward design techniques without introducing significant computational cost or expenses.

Due to budget limitations, we mostly aimed to show a proof-of-concept of our approach. All the contributions listed above held true for our tests, but without more experiments, the (2) and (3)' rd contributions above should be approached tentatively.

## 2. Background

### 2.1. Evolutionary Search with LLMs

Evolutionary search algorithms, drawing inspiration from biological evolution, involve the generation of outputs by a generator, such as a LLM [5]. The generated outputs undergo evaluation, leading to feedback that informs subsequent iterations of output generation. This iterative process includes the generation of outliers, thereby mitigating the risk of the algorithm converging to a local optimum.

A recent study demonstrated notable success in leveraging Evolution with LLMs for the design of reward functions, incorporating textual feedback and information from the environment [7]. In the present research, we extend this approach by introducing an additional modality of feedback—visual feedback—into the evolutionary process.

### 2.2. Multimodal Large Language Models (MLLMs)

While language is undoubtedly a crucial facet of human intelligence, our perception of the world extends beyond words. Humans perceive the world through various modalities, and each contributes to a comprehensive understanding of our surroundings. As LLMs have demonstrated exceptional proficiency in processing textual data, achieving notable successes on common benchmarks [10, 12], recent research have sought to expand the capabilities of these models by integrating other modalities, such as images, videos, and audios into LLMs. Many MLLMs have been proposed, including Llava [6], Flamingo [1], and GPT-4 [10].

Recent work has utilized Vision Language Models, which, in the context of our research, serve the same function as MLLMs, as direct reward model by using the cosine similarity between a state's image representation and the natural language task description [11]. This approach is similar to what we are trying to accomplish in this research, but differs mainly because of the fact that we are making MLLM generate code as a reward function, instead of it being a reward model itself.

## 3. Methods

We incorporate several methodologies to enhance the efficacy of reward design and optimization in reinforcement learning, building upon the foundation laid by [7]. We have used the environment as context, evolutionary search, and reward reflection as our primary methods. We call the unification of these approaches EROM: **E**volutionary **R**eward Design and **O**ptimization with **M**ultimodal Large Language Models. Our primary innovations are in the environment as a context part, where we provide the MLLM with the idle image of an agent, and in the reward reflection process, where we introduce video feedback into the loop.

### 3.1. Environment as Context

The model needs to have an understanding of the environment to generate a task-specific reward design for that environment. To achieve this, we give the environment source code as context to the model [7]. This helps because providing the environment code gives the MLLM essential information about the variables used in the environment code and in what format we expect an output. Additionally, we augment the contextual information by presenting the MLLM with visual representations of the environment and agent. We believe this helps MLLM understand the environment's visual cues and agent characteristics.

### 3.2. Evolutionary Search

We employ Evolutionary search for the iterative refinement of reward design. Initially, the model generates random samples of reward candidates, which are then evaluated on the task, and the top performer is selected. Subsequently, both reward feedback and the top performers are collected and fed back into the model for further enhancement. This iterative process is crucial, as evidenced by studies on LLMs demonstrating their capacity for self-improvement over time [8]. Moreover, this approach aligns with human intuition, as trial-and-error is a common strategy employed in the design of reward functions [3].

### 3.3. Reward Reflection

Previous studies utilizing LLMs to generate reward samples have primarily relied on textual feedback provided by the environment for evolutionary search [7]. However, capturing the visual behavior of an agent can also yield valuable insights into necessary adaptations. For instance, visual feedback can aid in identifying instances of reward hacking or pinpointing areas where the agent is not performing as intended. To address this, following the initial iteration of reward sampling, each reward function is individually tested, and both textual feedback from the environment and video recordings of the agent's performance are collected. Subsequently, for the subsequent iteration of evolution, the

MLLM is provided with the code of the best-performing reward function, along with its numerical and video feedback gathered during training. The MLLM then reasons over this information to iteratively design improved reward functions. Through this process of reward reflection, the accuracy of designed rewards consistently improves, leading to notable outcomes in our experiments.

## 4. Experiments

### 4.1. Baselines

In this subsection, we provide an overview of the critical components: simulation environment and the MLLM chosen for our research. We also describe a method that we will compare our method against.

#### 4.1.1 Environment

IsaacGYM [9] is a GPU-Accelerated Physics Simulation for robotics tasks. It enables hundreds of trainings to run at the same time, thus making it faster to conduct experiments. Also, we can capture videos during training, which is a prerequisite for our experiment. We picked humanoid and ant agents on two different tasks for our experiments on this simulator. The reason for selecting these agents was the GPU memory limit of our hardware.

#### 4.1.2 Multimodal Large Language Model

GPT-4V(Vision) [10] is a MLLM that can take both visual and textual input. Its multimodal capabilities will allow it to reason over videos and images, and its natural language and programming capabilities will allow it to understand tasks and generate reward functions as Python codes, making it suitable to use in our experiments.

#### 4.1.3 Eureka Method

Evolution-driven Universal Reward Kit for Agents (Eureka) [7] is a method that inspired us and the method that we built upon. The Eureka method involves providing the environment source code as context, evolutionary search to improve rewards, and using reward reflection. The only difference we made in our method is that we added visuals to the feedback loop and the environment as context part. We used very similar prompts to those of Eureka, with only minor changes indicating to the MLMM that we have added visuals. Also, Eureka has been shown to outperform 83% of human-expert-designed reward functions, which makes being able to outperform it a remarkable achievement.

### 4.2. Experimental Setup

We conducted three different tests to evaluate the effectiveness of our approach. Following the experiments originally described in the Eureka paper, we ran both EROM and Eureka for five iterations, generating 8 samples in each iteration. Due to the stochastic nature of MLLMs, when none of the codes worked in the first iteration, we reran it until at least one worked, resulting in guaranteed four rounds of feedback. We refer to this as "general testing" in the results subsection of our research.

We separately assessed the importance of providing an image of an agent in the first generation. We ran both EROM and Eureka for one iteration, generating 32 samples. We have increased the sample size to have more examples to lower the chance factors that could effect the results. We refer to this as "Image Testing" in the results subsection of our research.

We also separately assessed the importance of providing video during the feedback loop by providing the MLLM with the same reward codes generated in another iteration: one with only numerical feedback and the other with video feedback alongside numerical feedback. We generated 32 samples for both methods and compared them. We refer to this as "Video Testing" in the results subsection of our research.

Unless otherwise specified, when making experiments with EROM method, we provided the MLLM with a one-minute video of the agents training on the best policy generated during the training process (divided into 200 frames due to the context length of GPT-4V). In the reward sampling process, we trained the ant agent for 1500 epochs and the humanoid agent for 1000 epochs. In each training, the environment size was set to default for both agents. Each reward that achieved the best success rate in the initial training process was chosen to seed the next generation. We refer to the success rate obtained by reward functions in the initial iteration as "training-success" in the rest of the research. We evaluated the final best reward by retraining it over 5 different seeds and taking the average. We refer to this average as "average success."

### 4.3. Results

All the "average-success" results can be found in Tab. 1. Firstly, we observed that our method performed better on general testing, where we ran both codes for 5 iterations with 8 samples generated in each iteration. On ant and humanoid agents, EROM achieved an average-success rate of 7.27 and 5.26, while Eureka achieved an average-success rate of 3.68 and 4.21, respectively. We have also plotted the difference between EROM and Eureka over the "training-success" of each iteration on Fig. 1, Fig. 2. These graphs effectively demonstrate the effectiveness of evolutionary search for both methods, as well as the value of video feedback and providing the image of the agent.

Secondly, to test the importance of providing the image of an agent in the first generation, we generated 32 samples

Table 1. Average Success Rates

| Test Type | Ant-EROM | Ant-Eureka | Humanoid-EROM | Humanoid-Eureka |
|---|---|---|---|---|
| General Testing | $7.27\|0.36\sigma$ | $3.68\|0.71\sigma$ | $5.26\|0.29\sigma$ | $4.21\|0.53\sigma$ |
| Video Testing | $6.13\|0.95\sigma$ | $3.38\|0.39\sigma$ | $5.42\|0.27\sigma$ | $4.81\|0.70\sigma$ |
| Image Testing | $6.38\|1.89\sigma$ | $1.76\|0.87\sigma$ | $3.17\|0.30\sigma$ | $5.33\|0.39\sigma$ |

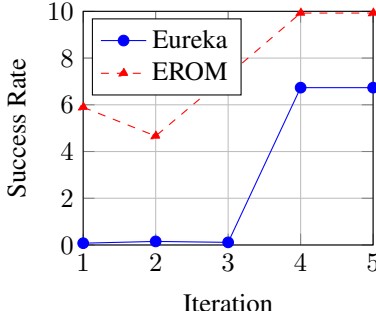

Figure 1. Comparison of success rates in General Testing on Ant agent.

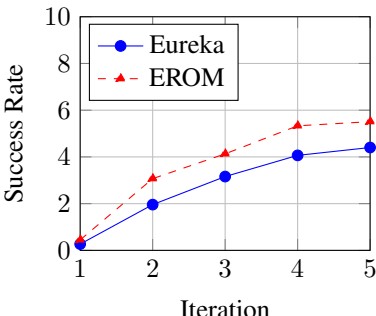

Figure 2. Comparison of success rates in General Testing on Humanoid agent.

using each method to increase the sample size and obtain a better average. As shown in Tab. 1, providing an image has shown to increase the average success rate for the ant agent, but not for the humanoid agent.

Lastly, by seeding the MLLM with the same reward functions and reward reflection, one with video and the other with only numerical feedback, we generated 32 samples with each method. We observed that providing the video also improved the average success for both of the agents.

### 4.4. Discussion

Our developed method, EROM, which aims to introduce multimodality into the reward design process, has shown success over previous applications that only utilize text. By observing average success rates on agents in our experiments specifically designed to assess the importance of providing an image of the agent, it can be concluded that providing the image of the agent can indeed be helpful, but not always. Secondly, we can see that introducing video in the feedback loop is beneficial, as our experiments have shown that, with the same seeds, but one with video feedback alongside numerical feedback, versus one with only numerical feedback, our approach performed better. Lastly, looking at the general experiments, we can see that performing evolutionary search with the EROM method is more effective than with the Eureka method in our experiments.

That said, our work has some limitations. Firstly, since we utilized GPT-4V [10] in our experiments, results largely depend on its capabilities. Additionally, the real-world applicability of our method may not match its success in online simulation environments, given the inherent complexity of real-world scenarios compared to simulations. Moreover, due to limitations in GPU memory, our experimentation was confined to only two agents in IsaacGYM. Expanding our tests to encompass a broader range of agents and environments would provide a more comprehensive assessment of our approach's generalization and efficacy.

## 5. Conclusion

Designing effective reward functions is a task that requires expertise and time. Recent researchers have sought to address this problem by utilizing LLMs to generate reward functions by taking the environment as context, employing evolutionary search, and utilizing reward reflection [7]. However, they have only used numerical feedback and textual information for reward sampling and the reward reflection process. In this work, we address this limitation by incorporating videos of agents in training and their idle images into the evolutionary process with the help of MLLMs. Our aim is to enhance the success rate of previous methodology, which have already outperformed 83% [7] of human experts in their focused tasks. Experiments conducted with two agents across two tasks have indicated that our approach is more effective than solely utilizing textual information.

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
