# OpenReview forum: "Evolutionary Reward Design and Optimization with Multimodal Large Language Models"
_thecvf.com/CVPR/2024/Workshop/VLADR — VLADR 2024 Poster_

### Official Review · Reviewer_9tSg · 2024-04-20
**Good exploration on VLLM reward model**

**Rating:** 6
**Confidence:** 3

**Review:**

The paper presents a novel approach, EROM, for designing reward functions in RL using Multimodal LLMs. The authors argue that previous methods have been limited to textual and numerical data and propose the incorporation of visual data, such as images and videos, to enhance the comprehension of the model and improve the accuracy of the generated reward functions. To further improve the paper, I suggest comparing it with more baseline models and conducting more experiments.

---

### Decision · Program_Chairs · 2024-04-22

Accept (Poster)